# The Influence of Rotational Length, along with Pre- and Post-Grazing Measures on Nutritional Composition of Pasture during Winter and Spring on New Zealand Dairy Farms

**DOI:** 10.3390/ani12151934

**Published:** 2022-07-29

**Authors:** Sagara N. Kumara, Tim J. Parkinson, Richard Laven, Daniel J. Donaghy

**Affiliations:** 1School of Agriculture and Environment, Massey University, Private Bag 11-222, Palmerston North 4410, New Zealand; d.j.donaghy@massey.ac.nz; 2Department of Farm Animal Production and Health, Faculty of Veterinary Medicine and Animal Science, University of Peradeniya, Kandy 20400, Sri Lanka; 3School of Veterinary Science, Massey University, Palmerston North 4410, New Zealand; t.j.parkinson@massey.ac.nz (T.J.P.); r.laven@massey.ac.nz (R.L.)

**Keywords:** nutritional composition, pasture, rotational grazing, energy, protein

## Abstract

**Simple Summary:**

Inconsistencies in the quality (nutritional composition) of pasture herbage can potentially affect the lactational performance of dairy cows. The nutritional composition of ryegrass–white clover pastures was grazed at different rotation lengths in New Zealand, on the hypothesis that longer rotations lengths would result in more mature swards, with lower quality herbage. Grazing rotation length, leaf regrowth stage at grazing, and pre-grazing DM yield during winter and spring had significant effects on the pasture herbage quality. In New Zealand, the length of the grazing rotation is adjusted based on the pasture availability, seasonal variations, and growth rate of different pastures. This information can be helpful for dairy farmers to manipulate pasture management to optimise the cows’ performance.

**Abstract:**

The quality of ryegrass–clover pasture was investigated between August (winter: start of calving) and November (spring: end of breeding) on pasture-based dairy farms (>85% of total feed from pasture) that had short (*n* = 2, Farms A and B; winter ~30 days, spring ~20–25 days) or long (*n* = 2, Farms C and D; winter ~35 days, spring ~25–30 days) grazing rotations to determine whether quality was affected by grazing rotation length (RT). Weekly assessments of pasture growth and herbage quality were made using a standardised electronic rising plate meter, and near-infrared spectroscopy, respectively. Data were subjected to repeated measure mixed model analysis, in which herbage quality was the outcome variable. The highest pre-grazing dry matter (PGDM) and height, post-grazing dry matter (DM) and height, and number of live leaves per tiller (leaf regrowth stage, LS) were present in late spring. Neutral detergent fibre (NDF), acid detergent fibre (ADF), metabolisable energy (ME), and organic matter digestibility (OMD) were positively correlated to each other (r^2^ ≥ 0.8) whilst ADF and lipid, and ADF and OMD were negatively correlated (r^2^ ≥ −0.8; *p* < 0.01). Metabolisable energy content was negatively correlated with ADF and NDF (r^2^ = −0.7, −0.8, respectively), and was inversely related to PGDM. Metabolisable energy was higher (*p* < 0.05) in farms with shorter (overall mean: 11.2 MJ/kg DM) than longer (10.9 MJ/kg DM) RT. Crude protein was also inversely related to PGDM and was higher with shorter (23.2% DM) than longer (18.3% DM; *p* < 0.05) RT. Pre-grazing DM affected the amount of pasture that was grazed and, hence, the amount of DM remaining after grazing (post-grazing DM or residual), so that PGDM was correlated with post-grazing height and residual DM (r^2^ = 0.88 and 0.51, respectively; both *p* < 0.001). In conclusion, RT, LS, and PGDM during winter and spring influenced the herbage quality, therefore, better management of pastures may enhance the productivity of dairy cows.

## 1. Introduction

Pasture herbage quality and pasture mass (dry matter (DM), kg/ha) are key factors when calculating the nutritional requirements of dairy cows in pasture-based dairy production systems [1,2,3]. In New Zealand, herbage quality and pasture mass both vary seasonally [4] and are affected by rotational grazing management [5], soil condition/nutrients [6,7], irrigation [7], and fertiliser application [8]. Grazing management has a significant effect on herbage quality through influencing seasonal productivity, digestibility, protein/energy balance, and level of rumen undegradable protein [9]. Herbage quality is therefore of high significance in pasture-based dairy production, as it critically affects the DM intake of cows, and consequent milk production, animal health, and reproduction.

In New Zealand, low- and medium-input farming systems, where pasture provides more than 80% of dry matter of the total diet, are common [10,11]. Perennial ryegrass (*Lolium perenne*) and white clover (*Trifolium repens*) are the predominant pasture types for more than 90% of dairy farms [12,13], even in those whose total reliance upon pasture is less than 80% of the total diet.

Adjusting the grazing rotation length (RT) according to pasture growth and stocking rate is important to maximise pasture utilisation [14,15], allow post-grazing targets to be met [16], and minimise any physical damage to the paddock and/or pasture [17]. The stocking rate can alter through the season as pasture growth and, hence, pasture availability for grazing stock varies with climatic conditions [18,19,20]. For example, low, moderate, and high pasture availability are prominent during winter, summer, and spring, respectively, and stocking rates are commonly changed between seasons to maintain an effective supply of pasture to grazing stock [21].

Decades of breeding and selection have improved the digestibility, DM yield, non-structural carbohydrate content, and condensed tannin levels of forage species [9], all of which can have a positive influence on milk production. However, long RT can result in feeding mature pasture with high fibre content, which leads to reduced digestibility and lower nutritive value (e.g., lower levels of metabolisable energy (ME) and crude protein (CP)), resulting in lower DM intake of dairy cows [22].

The present study reports on the variations in the herbage quality (nutrient profiles) of pre-grazing herbage samples collected from four spring-calving dairy farms over a period of four months, covering calving to the end of the first six weeks of the breeding program.

## 2. Materials and Methods

An observational study was conducted in low-input dairy farms (*n* = 4; nominated with capital letters from A to D) in the Manawatu region of the North Island of New Zealand. Farms were managed under Production System 1 or 2 [23], where pasture and supplements (imported feeds) were >85% and ~10% of the final diet, respectively. Dairy farms were chosen in consultation with local veterinarians and consultants who worked with a range of clients and who were asked to nominate farmers who had different grazing management approaches (RT, pre- and post-grazing management). Therefore, Farms A and B were selected as managing short RT, and Farms C and D were selected as managing long RT. The study was conducted between August (start of calving) and the end of November 2019 (end of first six weeks of breeding). Farms A, B, C, and D had effective land areas of 257 ha, 222 ha, 336 ha, and 68 ha, respectively, and were all stocked at 3 and 3.5 cows/hectare during winter and spring, respectively. Rotational grazing was practiced at all farms and RT are shown in Table 1. The grazing time was 18–20 h/day.

### 2.1. Pasture Types and Rotation Management

The predominant pasture was perennial ryegrass–white clover, however, Farm D also managed tall fescue (*Festuca arundinacea*)–white clover pasture (represented ~40% of grazing paddocks of the farm). Grazing rotation management at each farm was matched to the expected seasonal daily pasture growth rates, with faster rotations matching faster growth rates (e.g., spring), and slower rotations matching slower growth rates (e.g., winter).

### 2.2. Pasture Measurements

Pre-grazing pasture masses (PGDM; kg DM/ha) and pre-grazing pasture height (PGH; cm) were measured weekly using a standardised electronic rising plate meter (Tru-Test EC-10, Auckland, New Zealand), with more than 50 readings taken following a ‘W’ or ‘S’ pattern across the grazing paddock to cover a representative area. The plate meter was programmed with the following calibration equation [24,25]:Pasture mass (kg DM/ha) = (140 × mean compressed height) + 500

Farm maps were used to identify the paddock numbers and calculate the grazing intervals. On each weekly visit to each farm, a paddock that was about to be grazed was selected for pre-grazing measurements, and a paddock that had been grazed a day previously was selected for the post-grazing pasture measurements. Additionally, the perennial ryegrass leaf regrowth stage (LS; 10 measurements/paddock) was measured for each pre-grazing paddock, following the method outlined in McCarthy et al. [26].

### 2.3. Pasture Sampling and Processing

Pre-grazing pasture samples (*n* = 112, 28 samples/farm) were collected from >25 random locations in the paddock in duplicate using the hand-plucking method [27]. There were >40 paddocks in each farm. Samples were collected and placed polythene bags and transferred to the laboratory within 30–60 min. Each pasture sample was thoroughly mixed, and representative sub-samples (*n* = 4) were taken. Sub-samples were dried at 60–65 °C for between 48 and 72 h in a forced-draught oven until a constant mass was achieved. An electric balance was used to measure the mass of pasture before and after oven drying to determine the DM %. The dried samples were ground and sieved (1 mm) for chemical composition analysis.

### 2.4. In Vitro Analysis

The chemical composition of the pasture samples was assessed using near-infrared spectrophotometry (NIRS) [28]. The DM, CP, acid detergent fibre (ADF), neutral detergent fibre (NDF), and ash content were estimated using the methods of the Association of Official Agricultural Chemists (AOAC, Rockville, MD, USA) 930.16, AOAC 968.06, AOAC 973.18, AOAC 2002.04, and AOAC 942.05, respectively. Lipid and non-structural carbohydrate (NSC) contents were also measured by NIRS. Pasture samples were analysed for in vitro organic matter digestibility (OMD) and digestible organic matter digestibility (DOMD) according to Roughan and Holland [29,30]. The ME content of pastures was calculated using DOMD (0.163 × DOMD MJ/kg DM) [29,31].

### 2.5. Statistical Analysis

Data were tested for normal distribution using the Kolmogorov–Smirnov test and q–q plots. Log_(ln)_ transformation was undertaken where data were not normally distributed. Scatter plot analyses of herbage-related variables were used to visualise the relationships among variables. Correlation analyses were used to determine the strength and directions of relationships between variables. Comparisons of each nutritional component were undertaken both within and across months during late winter to the end of spring.

Data were thereafter subjected to analysis by a repeated measure mixed model in which herbage quality parameters (CP, NDF, ADF, lipid, NSC, ash, OMD, ME) were the outcome variables. The categorical predictor variable was the farm, and time (week) within farms were the repeated measures. The analyses were undertaken using the PROC MIXED function of SAS.

The linear mixed model developed was:$$y_{ij}=\mu +F_{i}+Time_{j}+F_{i}\times Time_{j}+e_{ij}$$
where *y* is the pasture DM or herbage quality parameter (CP, NDF, ADF, lipid, NSC, ash, OMD, ME); μ is the mean; *F* is the farm; *Time* is time periods during winter and spring, and e is the random error term.

Furthermore, linear regression models were applied to determine the association between pasture DM and herbage quality parameters (CP, NDF, ADF, lipid, NSC, ash, OMD, ME) as the outcome variables and PGDM yield, PGH, RT, and LS (number of live leaves per tiller) as the explanatory individual variables. The analyses were undertaken using the PROC MIXED function of SAS.

The linear mixed model developed was:$$y_{ijkl}=\mu +PGDM_{i}+PGH_{j}+RT_{k}+LS_{l}+PGDM_{i}\times RT_{k}+GH_{j}\times RT_{k}+e_{ijkl}$$
where *y* is the pasture DM or herbage quality parameter (CP, NDF, ADF, lipid, NSC, ash, OMD, ME); μ is the mean; e is the random error term and other factors as shown in Table 2.

Regression models were developed using 95% confidence intervals and 0.05 error. All analyses were undertaken using Statistical Analysis Software (SAS 9.4, SAS Institute Inc., Cary, NC, USA). Pearson and Spearman correlations were used to estimate the correlation of coefficients (r^2^) of the normally distributed and non-normally distributed data, respectively.

## 3. Results

The herbage quality values are summarised in Table 3. Compared to reference values [4,23,32], the average herbage quality values were at acceptable levels. The coefficient of variation between months was highest for NSC (13%), followed by ADF (11%). Ash, ADF, and NDF contents were significantly (*p* < 0.05) higher during the spring months of October and November than at other times, with CP showing a similar, non-significant trend. Lipid and ME contents were lower (*p* < 0.05) during October and November than at other times.

The NDF data separated into two homogenous subsets so that the mean NDF during August, October, and November were not different (*p* > 0.05) from each other (lowest mean difference August and October: −1.54% DM (adj. 95% CI: −3.16 to 0.06% DM)), but were higher than the mean NDF during September (lowest mean difference September and October: −2.17% DM (adj. 95% CI: −3.67 to −0.68% DM)). The mean NDF during August, September, and November were separated into another two homogenous sub-subjects so that the mean NDF during August and September were not different (*p* > 0.05) from each other (mean difference: 0.62% DM (adj. 95% CI: −0.98 to 2.24% DM)), but differed from the mean NDF during September to November (mean difference: −1.49% DM (adj. 95% CI: −3.1 to 0.11% DM)).

The ADF data separated into two homogenous subjects, so that the mean ADF during August and September were not different (*p* > 0.05) from each other (mean difference: 0.5% DM (adj. 95% CI: −0.62 to 1.63% DM)), but were lower than other months (lowest mean difference August and October: −1.13% DM (adj. 95% CI: −2.25 to −0.01% DM)). The mean ADF during October to November formed one homogenous subset (mean difference: −1.03% DM (adj. 95% CI: −2.15 to 0.09% DM)).

The ME data separated into two homogenous subjects, so that the mean ME during August and September were not different (*p* > 0.05) from each other (mean difference: 0.09 MJ/kg DM (adj. 95% CI: −0.15 to 0.33 MJ/kg DM)), but were higher than other months (largest mean difference August and October: 0.4 MJ/kg DM (adj. 95% CI: 0.16 to 0.64 MJ/kg DM)). The mean ME during October and November formed one homogenous subset (mean difference: −0.05 MJ/kg DM (adj. 95% CI: −0.29 to 0.18 MJ/kg DM)).

According to the Pearson correlation, strong positive relationships (r^2^ > 0.8) were observed between NDF and ADF, and between ME and OMD (Table 4). Similarly, strong negative relationships (r^2^ > 0.8) were observed between ADF and lipid, and between ADF and OMD. The ME content was negatively correlated with both ADF and NDF. The CP was positively correlated with ash, lipid, OMD, and ME; however, it was negatively correlated with DM, NSC, NDF, and ADF. Lipid contents were positively and negatively correlated with NDF and ME, respectively.

The highest PGDM and PGH, post-grazing DM and height, and LS were achieved in late spring (Table 5). There were significant differences observed between months for PGDM, PGH, post-grazing DM, residual/post-grazing height, and LS. The highest (32%) and lowest (8%) variabilities between months were observed for LS and post-grazing DM.

According to the Spearman correlation, the most significant positive relationships (r^2^ > 0.5) were observed between PGDM and residual height, and PGDM and post-grazing DM (Table 6). There were no negative relationships observed between parameters.

### Description of Models

The dry matter contents (Table 7, Figure 1) were affected by time period (week), farm, and by week × farm interaction (*p* < 0.05). There were interactions between PGDM and RT, and PGH and RT for the DM of pasture (*p* < 0.05, Table 7). The mean DM of RT2 (19.3% DM) was higher than that of RT1 (17.5% DM; *p* < 0.05). Similarly, the mean DM of PGDM1 (19.8% DM) was higher than that of PGDM2 (17.6% DM; *p* < 0.05). The highest DM contents were reported during time periods 4 and 11 at Farm D.

The crude protein contents (Table 7, Figure 2) were affected by the time period (week), farm, and by week × farm interaction (*p* < 0.01). There were interactions between PGDM and RT, and PGH and RT for the CP of pasture (*p* < 0.05, Table 7). The mean CP of RT1 (23.2% DM) was higher than that of RT2 (18.3% DM; *p* < 0.05). Similarly, the mean CP of PGDM1 (21.4% DM) was higher than PGDM2 (19.0% DM; *p* < 0.05).

The neutral detergent fibre contents (Table 7, Figure 3) were affected by time period (week), farm, and by week × farm interaction (*p* < 0.01). The NDF was influenced by RT and the mean NDF content of RT2 (44.6% DM) was higher than RT1 (42.0% DM).

The acid detergent fibre contents (Table 7, Figure 4) were affected by the time period (week), farm, and week × farm interaction (*p* < 0.01). The ADF was influenced by RT and LS (*p* < 0.05). The mean ADF content of RT2 (23.8% DM) was higher than RT1 (21.2% DM) and the mean ADF content of LS2 (23.7% DM) was higher than LS1 (21.3% DM).

The metabolisable energy contents (Table 7, Figure 5) were affected by the time period (week), farm, and by week × farm interaction (*p* < 0.01). The ME was influenced by PGDM, RT, and LS. The mean ME content of PGDM1 (11.2 MJ/kg DM) was higher than PGDM2 (10.9 MJ/kg DM). Similarly, the mean ME content of RT1 (11.2 MJ/kg DM) was higher than RT2 (10.9 MJ/kg DM), and the mean ME content of LS1 (11.2 MJ/kg DM) was also higher than LS2 (10.9 MJ/kg DM).

## 4. Discussion

The pasture herbage quality varied over the study period [from August (winter: start of calving) to November (spring: end of the first 6 weeks of the breeding period)], however, the herbage quality parameters were at acceptable levels (or at least ‘good’) according to the DairyNZ [16] guidelines. Herbage quality was significantly related to the RT, PGDM, PGH, and LS. Importantly, the ME and CP were lower when the PGDM was high (>3000 kg DM, Table 7) than when PGDM was low, and were also lower with longer rather than shorter RT. Moreover, the NDF and ADF contents were also increased with longer RT. Common pasture management practice is that RT is adjusted, sometimes according to different LS or otherwise based on pasture growth rates, to maintain the optimum pasture quality and quantity [22,33]. The leaf regrowth stage is primarily affected by temperature, and, to a lesser extent, soil moisture [33,34,35], which can be monitored weekly or monthly. All dairies selected in the current study were managed under the optimal LS stage (2–3 live leaves/tiller) [34] during late winter and early spring; however, LS went above the optimum (>3 leaves/tiller) during the period in late spring, where the maximum pasture growth occurred [36]. When LS increases beyond the optimum, dying leaves increase fibre levels and this, along with advanced reproductive development, decrease the digestibility [33,34]. Therefore, harvesting a surplus of grass for conservation at peak growth is essential to maximise pasture utilisation and quality [36,37,38,39]. In agreement with results from studies by Fulkerson and Donaghy [34] and Turner et al. [40], the present study also found that as the LS increased above 3, both the ME and CP contents were lower. Therefore, the optimum LS during the spring season and adjusting/lowering the RT accordingly may directly enhance the pasture quality [41] by decreasing the fibre content and increasing the digestibility and nutrient density [42,43]. In that context, it was clear that the RT was immediately changed if a farmer considered that either the LS and/or PGDM were unsatisfactory. Hence, these results highlight the adaptability of ryegrass–clover pastures [44,45], where the quality does not exhibit drastic changes between seasons, so that the ME contents of pasture generally meet the average requirements for lactating dairy cows (~11.0 MJ/kg DM).

The positive and negative correlations of the herbage quality traits (CP, ME, NDF and ADF) were similar to the published data from New Zealand [4,46,47]. However, the relationships between CP and both NDF and ADF were stronger (r^2^ > 0.4) than in the previously published data (r^2^~0.2–0.3). Metabolisable energy and CP were negatively correlated with both NDF and ADF, confirming that pasture maturity or longer RT lowers the herbage quality. The negative relationship between ME and NDF in the present study (r^2^ = −0.72) was less strong than that between ME and ADF (r^2^ = −0.8) (see Table 4), which raises an important question of whether the cellulose and lignin fractions of fibre (as measured by ADF) had a greater effect upon the pasture ME contents [46,48] than the hemicellulose fraction, which is reflected in the NDF content [49]. In support of this, the inverse relationship between ADF and OMD (r^2^ = −0.81) was stronger than that between NDF and OMD (r^2^ = −0.59). Furthermore, OMD was highly correlated with CP (r^2^ = 0.49), as Roche et al. [46] also noted. These authors considered that this occurred because the digestibility and CP contents were both greater under the optimum (2–3 leaves/tiller) rotational grazing management.

The contents of CP were positively correlated with lipid (r^2^ = 0.43) and negatively with NSC (r^2^ = −0.69), similar to the results published by Roche et al. [46] but higher than those published (r^2^ = 0.27 and r^2^ = −0.14, respectively) by Machado et al. [4]. Fatty acid synthesis takes place within the chloroplasts of plant cells, where more plant protein is also stored as the ribulose biphosphate carboxylase (rubisco) enzyme: consequently, a greater number of chloroplasts in the plant cells is associated with a greater protein content and also lipid/fat production [46,50]. The negative relationship between CP and NSC is well-documented [4] and due first to the stimulatory effect of N on plant growth (which causes an increase in the herbage CP levels, but reduces NSC through increased growth and respiration), and second, to the fact that plants use water-soluble carbohydrates (WSC) to convert nitrates that are taken up by the roots into amino acids in the plant.

In the present study, the plant CP content was highest during spring, in which the weather conditions (temperature, light, and water) combined to increase plant growth and development [46,48]; however, the NSC contents were considerably lower during spring as plants produce more structural carbohydrate (cellulose and lignin) to provide rigidity for the plant during its rapid growth [51]. Therefore, the conversion of WSC to structural materials corresponds with decreased NSC and increased ADF and NDF, confirming a negative relationship between NSC and ADF and/or NDF.

The crude protein content exceeded the average requirements for lactating dairy cows (18% DM) [16], as also found by Moller [52]. Cressman et al. [53] and Forster et al. [54] stated that milk production increases when feeding high CP diets, but high CP content can negatively affect reproductive performance [55,56,57]. Kalscheur et al. [58] and Ji and Dann [59] considered that feeding high CP diets is essential to maximise milk protein yields, since DMI limits the meeting of metabolisable protein requirements during early lactation. Importantly, high CP intake increases ammonia production in the rumen and slows down ammonia passing into the blood [60,61,62]. Cows use energy [63] to convert the toxic ammonia to less-toxic urea in the liver [64,65,66]. Interestingly, cows that are provided pastures with high CP content appear to be able to quickly adapt to the consequential higher blood urea concentrations, apparently obviating most of the toxic effects of high blood ammonia that are reported in the literature [67].

In the current study, pasture residual height was positively correlated (r^2^ > 0.5) with PGDM. Residual height can, in turn, be affected by decisions relating to pasture allowance that are based upon the pre-grazing DM yield and/or stocking density/rate [38]. Residual height is also affected by the quality of the pasture, as the amount of the plant that remains after grazing increases when the pasture contains more stem and dead material [68,69]. According to the DairyNZ [16] guidelines, farmers should be able to manage residual heights to between 3.5 to 4 cm, equating to 1500–1600 kg DM/ha targeted post-grazing DM. In the current study, where average residual height and post-grazing DM exceeded the DairyNZ [16] guidelines, there can be a positive effect upon pasture regrowth, and negative effects upon pasture wastage and pasture intakes at the next grazing [33,70]. This appears to have been the case in the present study, inasmuch as although pre-grazing DM was maintained between 2800–3200 kg DM/ha during the study period (which was within the DairyNZ [16] guidelines), the high post-grazing DM implies that pasture allocation exceeded the requirements and, therefore, the stocking rates should have been increased for more effective grazing management.

The contents of NDF in the current study were higher (>40%) than some of the values previously reported in New Zealand dairy systems [4]. Dairy cow rations should have at least 25% of DM as NDF to ensure that cows maintain rumen function [71] and, in pastoral systems, there should be a minimum of 35% of DM as NDF to maximize voluntary feed intake [16]. Conversely, excess dietary NDF limits DMI due to slow passage through the rumen and intestines [72,73].

The current study showed that intensive rotational grazing is crucial for optimising pasture quality, and hence, sustaining the nutritional value of the cows’ diet [12]. Hodgson and Brookes [25] suggested that grazing pasture at appropriate time intervals prevents the development of structural tissue and slows the decline of digestible cell contents, resulting in increased (or, at least, sustained) ME and CP contents. Appropriate management of grazing rotations in relation to the given nutritional requirements of grazing stock, along with the implementation of appropriate stocking rates, are important to reduce pasture wastage and increase milk production [74].

## 5. Conclusions

Different grazing RT, leaf regrowth stage at grazing, and pre-grazing DM yield during winter and spring influenced the quality of the pasture. The length of the grazing rotation was adjusted based on the pasture availability, seasonal variations, and growth rate of different pastures. The NDF, ME, and CP are key nutritional factors that can be used to determine the quality of the pasture, therefore, more attention on pasture monitoring, fertilizer application, irrigation, and maintenance of appropriate stocking density may be helpful to optimise the quality of the pasture between seasons. Proper rotational grazing management improves the quality of the pasture, therefore, this information can be useful for dairy farmers to manipulate pasture rotations to optimise the performance of cows.

## Figures and Tables

**Figure 1 animals-12-01934-f001:**
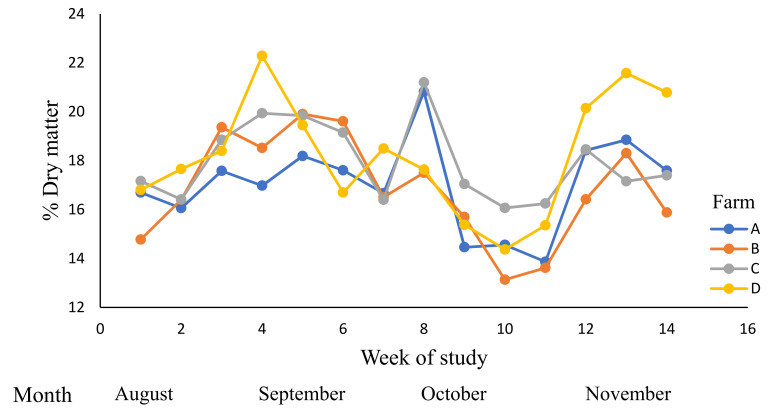
The variations in the dry matter (DM) content (%) of pasture across dairy farms (A–D) during winter and spring; 0–14 weeks (August to November 2019).

**Figure 2 animals-12-01934-f002:**
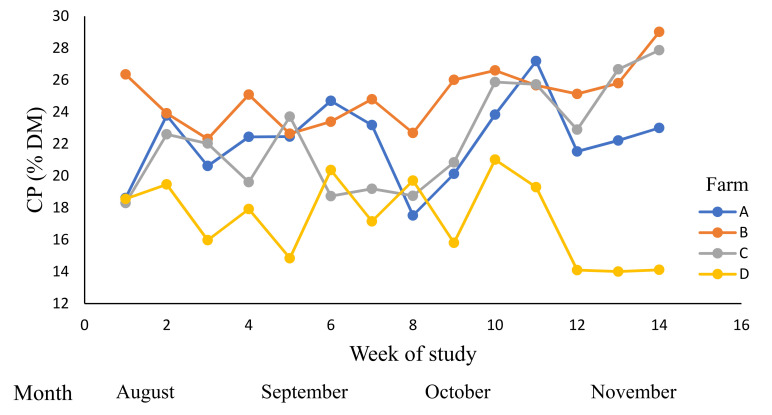
The variations in the crude protein (CP) content (% dry matter (DM)) across dairy farms (A–D) during winter and spring; 0–14 weeks (August to November 2019).

**Figure 3 animals-12-01934-f003:**
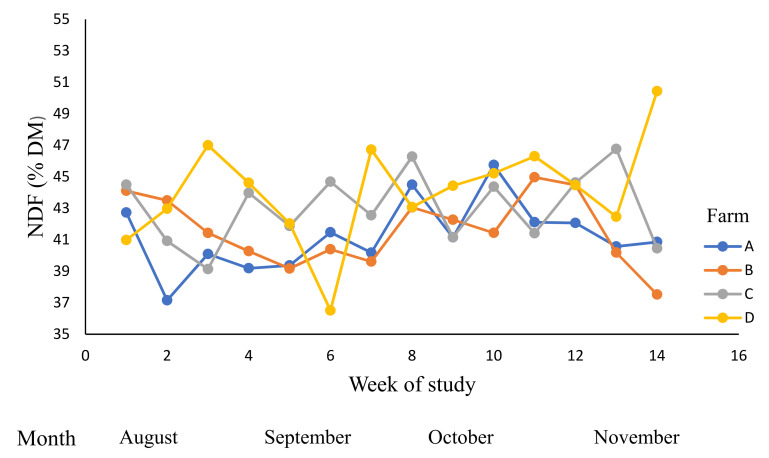
The variations in the neutral detergent fibre (NDF) contents (% dry matter (DM)) across dairy farms (A–D) during winter and spring; 0–14 weeks (August to November 2019).

**Figure 4 animals-12-01934-f004:**
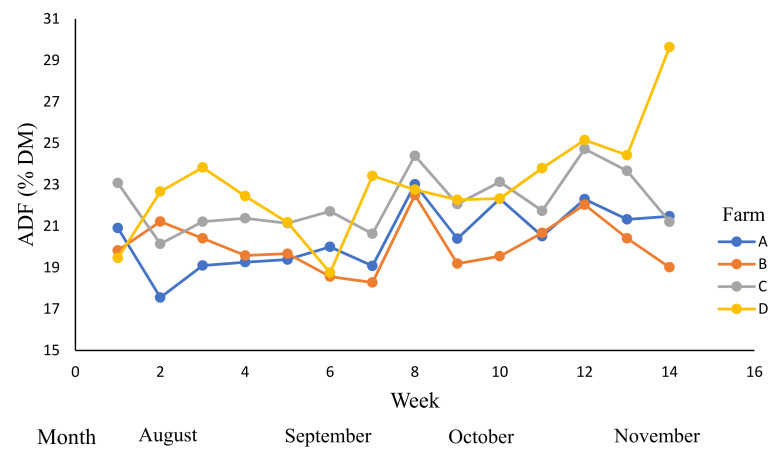
The variations in the acid detergent fibre (ADF) content (% dry matter (DM) across dairy farms (A–D) during winter and spring; 0–14 weeks (August to November 2019).

**Figure 5 animals-12-01934-f005:**
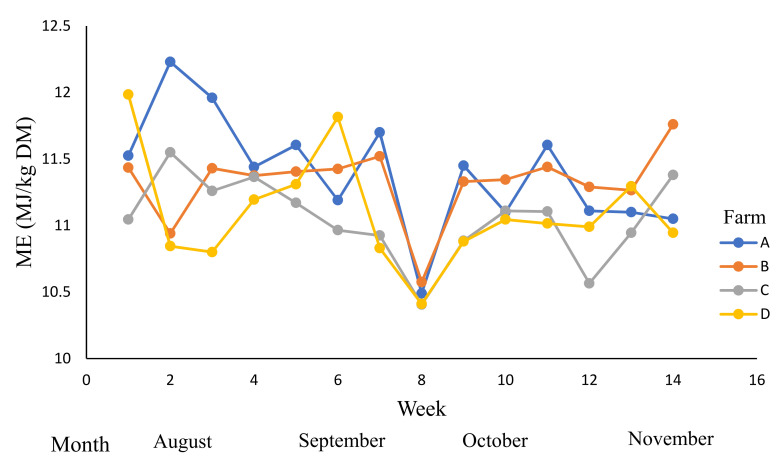
The variations in the metabolisable energy (ME) content (MJ kg dry matter (DM)) across dairy farms (A–D) during winter and spring; 0–14 weeks (August to November 2019).

**Table 1 animals-12-01934-t001:** The grazing rotation (days between grazing) of four dairy farms from August to November 2019.

Farm	Grazed Pasture	Grazing Rotation (Days)			
		Winter	Spring		
		August	September	October	November
A	Perennial ryegrass–white clover	30	20–25	20–25	20
B	Perennial ryegrass–white clover	30	25	20–25	20
C	Perennial ryegrass–white clover	35	30	25–30	25
D	Perennial ryegrass–white clover	35	30	30	25
	Tall fescue–white clover	40	35	35	30

**Table 2 animals-12-01934-t002:** The factors used in linear regression models to determine the association between pasture dry matter (DM) and herbage quality (nutrients).

Factor	Abbreviated Term	Ascribed Level	
		1	2
Pre-grazing DM (kg/ha)	PGDM	≤3000	>3000
Pre-grazing height (cm)	PGH	≤9	>9
Rotation length winter	RT	30 days	35 days
Rotation length spring	RT	20–25 days	25–30 days
Leaf regrowth stage at grazing	LS	≤3 leaves/tiller	>3 leaves/tiller

**Table 3 animals-12-01934-t003:** The herbage quality (nutritional composition; mean ± 95% confidence interval) over the entire study period (winter–spring) (linear mixed model analysis; *n* = 112).

	Norms *	Mean	August	September	October	November	r^2^
DM (DM %)	12–20	17.5	17.2 ^b^	18.51 ^a^	16.1 ^c^	18.4 ^a^	0.25
			(16.4–17.9)	(17.9–19.2)	(15.4–16.7)	(17.7–19.2)	
Ash (% DM)	8–12	9.95	9.44 ^b^	9.73 ^b^	10.3 ^a^	10.3 ^a^	0.13
			(9.05–9.82)	(9.4–10.1)	(9.97–10.6)	(9.91–10.7)	
Lipid (%) DM)	3–6	4.13	4.47 ^a^	4.44 ^a^	3.88 ^b^	3.7 ^b^	0.4
			(4.3–4.64)	(4.29–4.58)	(3.73–4.02)	(3.53–3.86)	
NSC (% DM)	7–25	11.8	12.1 ^a^	12.3 ^a^	10.7 ^b^	12.2 ^a^	0.06
			(11–13.3)	(11.3–13.3)	(9.7–11.7)	(11–13.3)	
CP (% DM)	15–30	21.7	21.3	21.6	22.2	22.3	0.02
			(19.3–22.5)	(19.9–22.6)	(20.9–23.7)	(20.6–23.8)	
NDF (% DM)	35–45	42.5	42.0 ^ac^	41.4 ^bc^	43.6 ^a^	42.9 ^ab^	0.08
			(40.8–43.3)	(40.4–42.5)	(42.5–44.6)	(41.7–44.1)	
ADF (% DM)	20–30	21.4	20.8 ^b^	20.3 ^b^	21.9 ^a^	22.9 ^a^	0.19
			(19.9–21.6)	(19.5–21)	(21.2–22.7)	(22.1–23.8)	
OMD (%) DM)	65–85	78.8	79.2 ^ab^	79.3 ^a^	78.5 ^ac^	77.9 ^bc^	0.04
			(78.2–80.3)	(78.4–80.3)	(77.6–79.5)	(76.8–79.0)	
ME (MJ/kg) DM)	10.5–12.5	11.2	11.4 ^a^	11.3 ^a^	11.0 ^b^	11.1 ^b^	0.13
			(11.2–11.6)	(11.2–11.5)	(10.9–11.2)	(10.9–11.2)	

NSC non-structural carbohydrates; CP crude protein; NDF neutral detergent fibre; ADF acid detergent fibre; OMD organic matter digestibility; ME metabolisable energy; SEM standard error of mean; Norms normal range. ^a–c^ Means with different superscripts within rows indicate significant differences between months (*p* < 0.05). * [4,22,29].

**Table 4 animals-12-01934-t004:** The correlation matrix for pasture dry matter (DM) and herbage quality nutrients (*n* = 112).

	DM	Ash	Lipid	NSC	CP	NDF	ADF	OMD	ME
DM	1								
Ash	−0.37 ****	1							
Lipid	−0.23 *	0.13	1						
NSC	0.37 ****	−0.62 ****	−0.06	1					
CP	−0.49 ****	0.79 ****	0.43 ****	−0.69 ****	1				
NDF	0.30 **	−0.05	−0.60 ****	−0.13	−0.41 ****	1			
ADF	0.35 ***	−0.21 *	−0.82 ****	0.05	−0.53 ****	0.84 ****	1		
OMD	−0.45 ****	0.33 ***	0.64 ****	−0.08	0.49 ****	−0.59 ****	−0.81 ****	1	
ME	−0.28 **	−0.01	0.64 ****	0.01	0.38 ****	−0.72 ****	−0.80 ****	0.83 ****	1

NSC non-structural carbohydrates; CP crude protein; NDF neutral detergent fibre; ADF acid detergent fibre; OMD organic matter digestibility; ME metabolisable energy; * *p* < 0.05, ** *p* < 0.01, *** *p* < 0.001, **** *p* < 0.0001.

**Table 5 animals-12-01934-t005:** The pre- and post-grazing pasture measurements (mean ± 95% confidence interval) (*n* = 112).

	Mean	August	September	October	November	SEM	r^2^
Pre-grazing DM (Kg DM/ha)	3001	3007 ^ab^	2912 ^b^	2951 ^b^	3179 ^a^	101	0.06
		(2850–3164)	(2776–3049)	(2815–3088)	(3021–3336)		
Post-grazing DM (Kg DM/ha)	1707	1707 ^ab^	1639 ^b^	1723 ^a^	1775 ^a^	36.3	0.11
		(1650–1764)	(1590–1689)	(1673–1772)	(1718–1832)		
Pre-grazing height (cm)	8.93	8.95 ^ab^	8.61 ^b^	8.75 ^b^	9.57 ^a^	0.66	0.1
		(8.39–9.52)	(8.13–9.1)	(8.27–9.24)	(9–10.13)		
Residual-height (cm)	4.31	4.31 ^ab^	4.07 ^b^	4.37 ^a^	4.56 ^a^	0.13	0.11
		(4.11–4.52)	(3.9–4.25)	(4.19–4.54)	(4.35–4.76)		
Leaf regrowth stage (no. of leaves per tiller)	2.29	1.90 ^c^	2.20 ^b^	2.28 ^b^	2.81 ^a^	0.10	0.41
		(1.74–2.05)	(2.07–2.34)	(2.15–2.41)	(2.66–2.97)		

DM—dry matter; SEM—standard error of mean. ^a,b,c^ Means with different superscripts within rows are significantly different (*p* < 0.05) from each other.

**Table 6 animals-12-01934-t006:** The correlation matrix for the pasture measurements (*n* = 112).

	Pre-Grazing DM	Post-Grazing DM	Pre-Grazing Height	Residual Height	Leaf Stage
Pre-grazing DM (kg DM/ha)	1				
Post-grazing DM (kg DM/ha)	0.51 ****	1			
Pre-grazing height (cm)	0.88 ****	0.16	1		
Residual height (cm)	0.51 ****	0.96 ****	0.16	1	
Leaf regrowth stage (leaves/tiller)	0.40 ****	0.26 ***	0.40 ****	0.26 **	1

DM—dry matter; ** *p* < 0.01, *** *p* < 0.001, **** *p* < 0.0001.

**Table 7 animals-12-01934-t007:** The least square means of main effects upon the pasture herbage quality (*n* = 112).

	PGDM (kg DM)			PGH (cm)			RT (Days)			LS			Interactions	
	≤3000	>3000	SEM	≤9	>9	SEM	1	2	SEM	≤3	>3	SEM	PGDM/RT	PGH/RT
DM (DM%)	19.8 ^a^*	17.6 ^b^*	0.7	18.1	18.8	0.7	17.5 ^b^*	19.3 ^a^*	0.7	17.6 ^b^*	19.2 ^a^*	0.6	S *	S *
Ash (% DM)	9.8	10.2	0.3	10.1	9.8	0.3	10.5 ^a^**	9.4 ^b^**	0.3	9.8	10.2	0.2	NS	NS
Lipid (% DM)	3.7	3.9	0.2	3.68	3.9	0.2	4.0 ^a^**	3.4 ^b^**	0.1	4.1 ^a^**	3.5 ^b^**	0.2	NS	NS
NSC (% DM)	11.8	11.9	0.9	11.4	12.2	1.0	11.5	12.1	0.8	11.9	11.5	0.9	NS	NS
CP (% DM)	21.4 ^a^*	19.0 ^b^*	1.1	21.7 ^a^*	19.4 ^b^*	1.2	22.2 ^a^**	18.3 ^b^**	1.1	21.1	20.4	0.7	S *	S **
NDF (% DM)	42.8	43.8	0.9	43.0	43.5	0.9	42.0 ^b^*	44.6 ^a^*	0.8	42.5	44.1	0.7	NS	NS
ADF (% DM)	22.1	22.9	0.6	22.3	22.7	0.6	21.2 ^b^**	23.8 ^a^**	0.5	21.3 ^b^**	23.7 ^a^**	0.6	NS	NS
OMD (% DM)	78.7 ^a^*	76.6 ^b^*	0.8	77.7	77.6	0.8	79.1 ^a^**	76.3 ^b^**	0.6	78.8 ^a^*	76.6 ^b^*	0.7	NS	NS
ME (MJ/kg DM)	11.2 ^a^*	10.9 ^b^*	0.1	11.1	11.0	0.1	11.2 ^a^*	10.9 ^b^*	0.1	11.2 ^a^*	10.9 ^b^*	0.1	NS	NS

^a,b^ Values with different superscript in the same row are different from each other (LSD, ** *p* < 0.001; * *p* < 0.05); the PGDM/RT interaction between pre-grazing dry matter and grazing rotation length; the PGH/RT interaction between pre-grazing height and grazing rotation length; LS—leaf regrowth stage at grazing; DM—dry matter; NSC—non-structural carbohydrates; CP—crude protein; NDF—neutral detergent fibre; ADF—acid detergent fibre; OMD—organic matter digestibility; ME—metabolisable energy; SEM—standard error of mean.

## Data Availability

Data is contained within the article.

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
