# Peer review of "The Influence of Rotational Length, along with Pre- and Post-Grazing Measures on Nutritional Composition of Pasture during Winter and Spring on New Zealand Dairy Farms"

_animals, 2022, doi:10.3390/ani12151934_

Round 1

Reviewer 1 Report

Dear Editor and Authors,

I send you my review about the article “The influence of rotational length, along with pre- and post-grazing measures on nutritional composition of pasture during winter and spring on New Zealand dairy farms”.

As reported in the aim, the main scope of the article was to study the variations in herbage quality of pre-grazing herbage samples collected from four spring-calving dairy farms over a period of four months, covering calving to the end of the first six weeks of the breeding program.

In my opinion, the article, although it result well written and it is sufficiently well structured, it need of some little change that I report below.

The introduction is a bit too concise and it does not adequately explain the originality and innovativeness of the research.

Therefore, in the introduction should be better explained the originality of this paper. To improve the originality of the article I suggest to the Authors, to reports the more recent article that have study the same aspects of their paper. After, they should stress, always in the introduction, the difference among this study and the others previously reported.

This operation has already been partially done, however, it should be completed.

Furthermore, to facilitate the understanding of the text by the readers the use of symbol in the text should be reduced as much as possible.

As examples, at line 81 the sentence “(n=4; nominated A-D)” should be replaced with the sentence “(n=4; nominated with capital letters from A to D)”. Again, at line 126, the symbol “>” should be replaced with “over”.

The results is well presented and they are well discussed, also in comparison to the data reported in the literature.

However, to facilitate the reading of the data shown in the table 3, 4, 5, 6 and 7 the number of samples or the trials should be reported in the tables or in their caption.

Finally, the conclusion resulted adequate to the data showed and to the aim of the research.

Best regards

Author Response

Dear Reviewer,

Thank you

Reviewer 2 Report

* The influence of rotational length, along with pre- and post-grazing measures on nutritional composition of pasture during winter and spring on New Zealand dairy farms

Dear authors,

Thank you very much for allow me to read this interesting manuscript. The text
is really pleasant to read and it's well written and structured.
This observational study describes associations between pasture quality and
rotation length, as well as pre- and post-grazing indicators of pasture biomass
and quality.

I think that the most interesting point in this study is the use of repeated
measures in time. That type of study almost always generate good quality data
and rich information about the biological behavior of grazing ecosystems.

I suggest you make the data available in order to make this research more
visible, improve the repeatability of the study and to facilitate future studies
(e.g., meta-analysis).

I offer the following comments and hope they can be useful to improve the
manuscript.

** Abstract

** Introduction

L 56:

- Is this value based on total dry-matter intake?

L 72:

- Please check this typo: there's an additional closing parentheses.

L 73-75:

- Consider to include any reference to grazing rotation length.

L 76-77:

- This information seems to be more suitable to be included in the Material and
  Methods section

** Materials and Methods
L 81:

- Should it be "observational study" instead of "observational studies"?

L 87:

- For how long did the cows stay in each paddock?

L 115:

- Did you report how many paddocks and how many samples were taken by season and
  farm?
  - I'm not sure if that information is clearly described in the manuscript.

L 153:

- Please remove the duplicated word.

L 167:

- There's no 'B0' in the formula.
- Why did you use a mixed-effects model to account for pseudoreplicated measures
  in the first model, but did not use the same approach in the second model?

** Results
L 142:
- Did you asses the goodness of fit of the models?
  - Please report that information

L 185-212:

- I think that statistical coefficients are easier to read when displayed
  in Tables or Figures. Figures are a great tool for showing time trends in
  data

L 217 (here and elsewhere):

- Do you think is it necessary to include both CI 95% and SEM?
- There are some typos: some parentheses are not closed
- I'm not sure if the meaning of 'Norms' is clearly recognized looking at the
  Table
- I suggest you include information about the statistical analyses used to model
  the data as well as the number of observations per variable and the coefficients
  for random effects when necessary

  L 227:

  - Please check this typo: a closing parentheses is missing

L 247:

- I suggest you reconsider the name of this Table. Try to make it more readable.
  Perhaps some specific, but valuable information can be moved into a footnote.
- How it should be interpreted the values used for 'rotation length' variable?
  Are those numbers a dummy variables?

L 260 (Figure 1 and elsewhere):

- This figure is a good option to show your results, however I wonder if it
  is really necessary to split the data by farms. Since your hypotheses are
  related to questions on associations between DM production, pasture heightm,
  rotation length and number of leaves, why not using those variables figures?
- In addition, the use of averages by and dispersion measures (i.e., CI, SEM)
  filtered by the variables of interest would allow the readers to better
  visualize the results and may help to reduce the length of the text in the
  Tables.

** Discussion

** Conclusions

L 406-407:

- I'm not sure if this is a good statement to be included in the Conclusions.
  The length of the grazing rotation was fixed in order to explore the
  associations between that variable and other variables.

L 412-414:

- This sentence makes sense, however it is not a conclusion obtained from your
  study. Please verify it and reword, if necessary.
- Please read all the conclusions again and try to adjust the text to be more
  focused on your hypotheses and study objectives

Author Response

Dear Reviewer,

Thank you.
